# Graphene Oxide Composite for Selective Recognition, Capturing, Photothermal Killing of Bacteria over Mammalian Cells

**DOI:** 10.3390/polym12051116

**Published:** 2020-05-13

**Authors:** Gang Ma, Junjie Qi, Qifan Cui, Xueying Bao, Dong Gao, Chengfen Xing

**Affiliations:** 1National-Local Joint Engineering Laboratory for Energy Conservation in Chemical Process Integration and Resources Utilization, School of Chemical Engineering and Technology, Hebei University of Technology, Tianjin 300131, China; magang0629@163.com; 2Key Laboratory of Hebei Province for Molecular Biophysics, Institute of Biophysics, Hebei University of Technology, Tianjin 300401, China; cuiqifan1228@163.com (Q.C.); XueyingBao2017@163.com (X.B.); gaodong@iccas.ac.cn (D.G.)

**Keywords:** photothermal therapy (PTT), graphene oxide (GO), selective recognition and capturing, antibacterial

## Abstract

The multifunctional photothermal therapy (PTT) platform with the ability to selectively kill bacteria over mammalian cells has received widespread attention recently. Herein, we prepared graphene oxide-amino(polyethyleneglycol) (GO-PEG-NH_2_) while using the hydrophobic interaction between heptadecyl end groups of 1,2-distearoyl-sn-glycero-3-phosphoethanolamine-N-[amino(polyethyleneglycol)] (DSPE-PEG-NH_2_) and graphene oxide (GO). Based on GO-PEG-NH_2_, the versatile PTT system was constructed with simultaneous selective recognition, capturing, and photothermal killing of bacteria. When the cells undergo bacterial infection, owing to the poly(ethylene glycol) (PEG) chains and positively charged amino groups, GO-PEG-NH_2_ can specifically recognize and capture bacteria in the presence of cells. Meanwhile, the stable photothermal performance of GO-PEG-NH_2_ enables the captured bacteria to be efficiently photothermally ablated upon the irradiation of 808 nm laser. Besides, the GO-PEG-NH_2_ is highly stable in various biological media and it exhibits low cytotoxicity, suggesting that it holds great promise for biological applications. This work provides new insight into graphene-based materials as a PTT agent for the development of new therapeutic platforms.

## 1. Introduction

The appearance of antibiotic resistance to pathogen bacteria is one of the major challenges in the global health field [1]. As an alternative approach, nanomaterials absorbing a specific wavelength of light have been exploited for the treatment of bacterial infections. Photothermal therapy (PTT) and photodynamic therapy (PDT) have attracted significant attention in recent years as photo–triggered antibacterial methods [2,3,4,5,6,7]. Unlike target-specific antibiotics, PTT and PDT are broad-spectrum pathogen killing methods that prevent the generation of specific resistant strains [8,9,10,11]. Recently, a great deal of pathogen killing systems has been explored, and multiantibiotic resistant bacteria can be easily killed by photo–triggered antibacterial methods [12,13,14,15]. However, the infection of mammalian hosts by pathogens often leads to the coexistence of both bacteria and cells. Hence, how to selectively kill bacteria in the presence of mammalian cells has become a research hotspot. Wang and co-workers have synthesized a water-soluble cationic poly(p-phenylene vinylene) derivative (PPV-1) containing Oligo(ethylene glycol) (OEG) side chain and quaternary ammonium (QA) groups, and pathogenic bacteria are killed in the presence of cells by the dark-toxic QA group and the light-toxic PPV-1 main backbone to produce Reactive Oxygen Species (ROS) under white light [16]. Liu and co-workers have designed a multifunctional aggregation-induced emission-zinc(II)-diamine (AIE-ZnDPA) probe while using a positively charged zinc(II)-difunctional amine (ZnDPA) as a ligand to selectively target negatively charged bacterial membranes via electrostatic interaction. The positively charged ZnDPA groups killed pathogenic bacteria to depolarize the bacterial membrane and phototoxicity through the generation of ROS [17]. The above strategies all utilize photosensitizers (PSs) to generate ROS upon light irradiation to kill bacteria. PTT is another alternative approach to treat against bacteria that usually can lead to thermal ablation of microorganisms and cells by using near infrared (NIR) light to transform optical energy into heat [4,8,18,19,20]. Besides, PTT also has the advantages of simplicity, non-invasiveness, safety, and strong tissue penetration [21]. Zhang and co-workers have designed a supramolecular complex (CPPDI) by a perylene diimide derivative (PPDI) and cucurbit [7] uril (CB [7]) through host-guest interactions. CPPDI could be triggered into CPPDI radical anions in situ by anaerobic bacteria (such as *E. coli*), which was used in photothermal therapy under near-infrared radiation in order to achieve selective killing of bacteria [22]. Although some progress has been made in photothermal materials with selective antibacterial capabilities, developing novel photothermal agents with excellent selectivity toward bacteria and cells is still imperative. Based on this conception, we try to use PTT to achieve the selective capture and killing of bacteria over mammalian cells. 

There are many types of photothermal conversion materials due to the tremendous potential of photothermal therapy applications [23,24], including precious metal nanomaterials [25], carbon nanomaterials [26,27], and transition metal dichalcogenides nanomaterials [28], have been explored for PTT in recent years. Graphene is a two-dimensional (2D) carbon nanomaterial that is composed of carbon atoms with sp^2^ hybrid orbital hexagonal honeycomb lattice [5,29,30]. graphene oxide (GO) and reduced graphene oxide (rGO) have been extensively used in various fields because of their impressive physical and chemical properties [31,32,33,34,35,36]. For example, the diverse defects and chemical functionalities on the graphene layers can catalyze reactions, while graphene can be used as a carrier for the catalytic reaction [37,38,39]. Besides, carbon nanomaterials such as graphene oxide have proven to be very promising nano weapons against multi-drug resistant bacteria [40,41]. Absorbing light irradiation can enhance the antibacterial properties of carbon nanomaterials against multidrug-resistant pathogens [42,43,44]. GO is an excellent PTT photothermal material with high photothermal conversion efficiency, high stability, and good biocompatibility [21,45,46,47,48,49].

In this work, we designed a DSPE-PEG-NH_2_ (one end of DSPE-PEG-NH_2_ is a hydrophobic structure 1,2-distearoyl-sn-glycero-3-phosphoethanolamine, and the other end is an amino-terminated PEG chain) modified GO composite (GO-PEG-NH_2_) [50,51,52]. The cell wall of bacteria is mainly made of peptidoglycan and acid polysaccharide, which exhibits a large amount of net negative charges on the membrane surface. Under physiological conditions, mammalian cell membranes are nearly neutral with only a small quantity of negative charge. Therefore, the amino group of GO-PEG-NH_2_ can be bound to a negatively charged bacterial membrane by electrostatic driving forces, and the presence of a PEG chain can also reduce the non-specific adsorption of cell surface proteins. The flexibility of the main polyether chain is responsible for these properties in this process [16,53]. At the same time, as a photothermal conversion material, GO can thermally ablate bacteria that are adsorbed on the surface of GO-PEG-NH_2_ composite under 808 nm laser irradiation.

The most attractive finding of this work is that we prepared GO-PEG-NH_2_ in an uncomplicated way to selectively identify and capture pathogenic bacteria and simultaneously kill pathogenic bacteria by photothermal. In addition, GO-PEG-NH_2_ is found to be a green broad-spectrum antibacterial material, with little bacterial resistance and a tolerable cytotoxic effect on mammalian cells.

## 2. Materials and Methods

### 2.1. Reagents and Equipment

Graphene oxide (GO, Product code G139803) was purchased from Shanghai Aladdin Biotechnology Co., Ltd. (Shanghai, China). 1,2-distearoyl-sn-glycero-3-phosphoethanolamine-N-[methoxy(polyethyleneglycol)-2000(ammonium salt) (DSPE-PEG2000-NH_2_) were purchased from Ananti Polar Lipids, Inc. (Beijing, China). Fetal bovine serum (FBS) was obtained from Sijiqing Biological Engineering Materials (Hangzhou, China). RPMR-1640 was purchased from Solarbio (Beijing, China). Isopropyl-beta-D-thiogalactopyranoside (IPTG) was purchased from Amresco Inc. (Houston, TX, USA). The human acute lymphoblastic leukemia cells (CCRF-CEM) were obtained from the Cell Bank of the Committee of Culture Collection of the Chinese Academy of Sciences (Shanghai, China).

A Malvern Nano-ZS90 (Malvern Inc., Malvern, UK) provided average particle sizes. The ultraviolet (UV)- visible light (vis) absorption values were taken on a Shimadzu UV-1800 UV-vis spectrophotometer (Shimadzu, Kyoto, Japan). The 808 nm laser was obtained from a Hi-Tech high power laser generator (Beijing, China). The absorbance for methylthia-zolyldiphenyl-tetrazolium bromide (MTT) analysis was taken on a SpectraMax i3 (Molecular Devices Inc., San Francisco, CA, USA) at a wavelength of 520 nm. Fluorescence imaging was recorded on a confocal laser scanning microscope (CLSM, Leica, Weztlar, Germany).

### 2.2. Synthesis of GO-PEG-NH_2_

GO was sonicated for 2 h with an ultrasonic cell disrupter, and then centrifuged at 15,000 rpm for 10 min. DSPE-PEG2000-NH_2_ (5 mg) and GO (2 mg/mL 0.5 mL) with a mass ratio of 5:1 were dispersed in 1 mL Tetrahydrofuran (THF) and sonicated for 30 min. The resulted material was added dropwise to 4 mL of ultrapure water and then stirred at 450 rpm for 12 h. The larger particle sizes of GO-PEG-NH_2_ were removed by centrifugation (3000 rpm for 10 min.). Small molecules were removed by dialysis, and unreacted starting materials were removed by ultrafiltration.

### 2.3. Preparation of Bacterial Solutions

A single colony of Escherichia coli (*E. coli*) on a solid Luria Broth (LB) agar plate was picked and transferred to 7 mL liquid LB culture medium and then grown overnight at 37 °C. A certain volume of the bacterial liquid was taken, and the bacteria were harvested by centrifugation (7500 rpm for 5 min.) and washed three times with phosphate-buffered saline (PBS, 10 mM, pH = 7.4). Finally, the obtained bacteria were resuspended in PBS and diluted to an optical density of 1.0 (OD_600_ = 1.0) at 600 nm. As for Staphylococcus aureus (*S. aureus*), the experimental conditions and operations were identical to that of *E. coli*.

### 2.4. Cell Culture

CCRF-CEM cells (suspension cells) were cultured in RPMI-1640 medium that was supplied with 15% fetal bovine serum (FBS). A certain amount of the culture solution was added to a 50 mL cell culture flask and then cultured at 37 °C in a humified atmosphere containing 5% CO_2_. CCRF-CEM should be mixed with the cell culture medium in the course of the passage, and half of the culture solution should be discarded, and fresh RPMI-1640 medium should be added. When the cells are in good condition, it requires replacing by liquid replacement every other day.

### 2.5. Cytotoxicity Assay

In this experiment, the cytotoxicity analysis of GO-PEG-NH_2_ was evaluated by the standard MTT assay. The CCRF-CEM cells were seeded in 96-well plates at a density of 1× 10^4^ ~3 × 10^4^ cells/well and then incubated overnight. Various concentrations of GO-PEG-NH_2_ in fresh culture medium were added into the cells. The cells were incubated at 37 °C for 24 h. The cells were washed with PBS. Unbound GO-PEG-NH_2_ was removed by centrifugation (300 rpm for 5 min.), and then 200 µL of MTT (0.5 mg/mL, dissolved in cell culture medium without bovine serum) was added to each well and incubated for 4 h at 37 °C. The supernatant was removed by centrifugation (300 rpm for 10 min.), and 150 µL of Dimethyl sulfoxide (DMSO) was added to each well and shaken in a shaker for 10 min. to dissolve the produced formazan. The absorbance values of the wells were read with a microplate reader at 520 nm. The untreated cells served as the control and their viability was set as 100%.

### 2.6. ζ Potential Measurements

CCRF-CEM, *E. coli* and *S. aureus* in PBS were incubated with 50 µg/mL GO-PEG-NH_2_ at 37 °C for 30 min., respectively. Unbound GO-PEG-NH_2_ was removed by centrifugation (500 rpm 5 min. for CCRF-CEM; 1500 rpm 5 min. for bacteria), and the bottoms pellet was resuspended in PBS. As a control, bacteria and cells without GO-PEG-NH_2_ were incubated under the same conditions. DLS measured the potential values of the cell suspension and the bacterial suspension.

### 2.7. Antibacterial Experiments

The antibacterial test was evaluated with the colony counting method [54,55]. *E. coli* and *S. aureus* bacterial suspensions with an OD_600_ of 1.0 were incubated with different concentrations of GO-PEG-NH_2_ under dark at 37 °C for 1 h. The supernatant was removed by centrifugation (1500 rpm for 5 min.), the sediment was collected, and the bacteria were resuspended in PBS. Subsequently, 200 µL of the bacterial suspension was added to a 96-well U-shaped cell culture plate and the bacterial suspension was irradiated with an 808 nm laser at an optical density of 1.5 W/cm^2^ for 5 min. After irradiation, the bacterial suspensions were serially diluted 1.0 to 1 × 10^4^ fold with PBS. A 10 µL portion of the diluted bacterial suspension was spread on the solid LB agar plate and then incubated at 37 °C for 12 h. The number of colonies was counted using the colony counting method and the bacteria survival rates were determined from CFU counting on the solid LB agar plate with the control under dark without GO-PEG-NH_2_ treatment. For the control under dark, the irradiation step was replaced by incubation under dark for 5 min.

### 2.8. Scanning Electron Microscopy (SEM) Characterization

SEM was used to observe the morphological changes of bacteria [16,17]. *E. coli* and *S. aureus* bacterial suspensions with an OD_600_ of 1.0 were incubated with GO-PEG-NH_2_ (50 µg/mL) under dark at 37 °C for 30 min, as described in antibacterial experiments. The concentration of GO-PEG-NH_2_ is determined by a standard curve (Appendix A). After irradiation, the bacteria were harvested by centrifugation, and immediately transferred to glutaraldehyde (2.5%) in PBS solution and fixed at 4 °C for 4 h. The bacteria were then collected by centrifugation and washed three times with sterile water. Dehydration was carried out sequentially with 10%, 30%, 50%, 70%, 90%, and 100% ethanol solution. Finally, the bacteria in 100% ethanol solution were added dropwise to clean silicon slices and then dried under vacuum overnight. The specimens were coated with platinum before SEM studies.

### 2.9. E. coli was Induced by IPTG to Express Recombinant Green Fluorescent Protein

A single colony of *E. coli* (Gram-negative) on a solid Luria Broth (LB) agar plate was picked and transferred to 10 mL liquid LB culture medium and grown at 37 °C for 20 h. Subsequently, IPTG (1 mM) was added to the *E. coli* solution in the log phase. After that, *E. coli* was continued at 37 °C for 4 h. A certain volume of the bacterial liquid was taken, and the bacteria were harvested by centrifugation (7500 rpm for 5 min.) and then washed three times with phosphate-buffered saline (PBS, 10 mM, pH = 7.4). Finally, the obtained bacteria were resuspended in PBS and then diluted to an optical density of 1.0 (OD_600_ = 1.0) at 600 nm.

### 2.10. Confocal Laser Scanning Microscopy (CLSM) Characterization

The prepared *E. coli* and CCRF-CEM cell solutions were incubated with GO-PEG-NH_2_ for 30 min. at 37 °C under the dark, respectively. Afterwards, 10 μL of the mixed suspension was added to the alcohol-treated glass slides and gently fixed with clean coverslips. Confocal laser scanning microscopy imaged the samples. The excitation wavelength is 488 nm.

For confocal images of the suspended cells and bacteria mixtures, the prepared *E. coli* and CCRF-CEM cell solutions were mixed and then incubated with 50 μg/mL GO-PEG-NH_2_ at 37 °C for 30 (and 60) min. under the dark. Individual aliquots of 10 μL of the prepared suspension were spotted on alcohol pretreated glass slides and then immobilized by the coverslips. Confocal laser scanning microscopy imaged the samples. The excitation wavelength is 488 nm. The fluorescence of *E. coli* is highlighted in green. 

## 3. Results and Discussion

### 3.1. Preparation and Mechanism of GO-PEG-NH_2_

Scheme 1 shows the mechanism for recognition, capturing, and photothermal killing of bacteria over mammalian cells. DSPE-PEG-NH_2_ can be tightly attached to GO through hydrophobic interactions between heptadecyl end groups and GO, yielding the surface of GO film by amino groups and PEG chains [26,50,56,57]. The membrane potential difference between bacteria and cells results in a competitive attraction of GO-PEG-NH_2_ to bacteria. The accumulation of bacteria on the surface of GO-PEG-NH_2_ by electrostatic interaction leads to the sedimentation of bacteria and GO-PEG-NH_2_. The GO converts light energy into heat upon irradiation under NIR light (808 nm) with the optical density of 1.5 W/cm^2^, which causes the thermal ablation of the bacteria.

### 3.2. Characterization of Structure and Photothermal Properties

The morphology of GO-PEG-NH_2_ was characterized by atomic force microscope (AFM), as shown in Figure 1a [39]. GO-PEG-NH_2_ was almost a single-layered sheet with an average topographic height of ~1.75 nm (Figure 1b). When compared to GO, the average topological height of GO-PEG-NH_2_ has increased by a certain amount (from ~1.37 nm to ~1.75 nm, Appendix A), indicating that many polymer chains have been wrapped around the surface of the GO sheets. It has been documented that the cell membrane of bacteria is effectively destroyed by direct contact between the bacteria and the very sharp edges of graphene, resulting in the inactivation of bacteria by grapheme [58,59]. The ultrasonic vibration and mechanical stirring may remove the sharp edges of GO during the preparation process [60]. The edges of GO sheets are very sharp, while the edges of GO-PEG-NH_2_ are smoother and rounder, which may enhance biocompatibility, as shown in the AFM charts. The more rounded edges reduce the physical damage to the cell membrane by GO-PEG-NH_2_. The SEM image of GO-PEG-NH_2_ exhibits a sheet-like structure with a smooth surface, small thickness, and small wrinkles at the edges, as illustrated in Appendix A. Dynamic light scattering (DLS) shows that the average hydrodynamic size of GO-PEG-NH_2_ was smaller than GO (from 956.90 ± 17.01 nm to 459.03 ± 10.54 nm, Appendix A), which might be caused by ultrasonic vibration during the preparation process. The physicochemical characteristics of graphene, such as the lateral size, surface charges, and surface functional groups, can have a significant impact on its biocompatibility. The reduction of lateral size can reduce the threat of GO-PEG-NH_2_ to the cells in a mixed state to a certain extent [12,61]. The potential of GO-PEG-NH_2_ is -4.27 ± 0.52 mV, which became more cationic after the functionalization of DSPE-PEG-NH_2_ (Table 1), indicating that positively charged amino groups successfully bind to the surface of GO-PEG-NH_2_. As an experiment to verify the stability of GO-PEG-NH_2_, it was incubated in various biological media (water, PBS, RPMI-1640 medium, and fetal bovine serum) for 24 h (Appendix A). The introduction of the PEG chain increases the hydrophilicity of GO-PEG-NH_2_. Besides, the relatively small particle size of the GO-PEG-NH_2_ composite also gives it better stability. The stability experiment shows the enormous potential of GO-PEG-NH_2_ in biological applications. 

The absorption spectrum of GO-PEG-NH_2_ shows a broad absorption band from the ultraviolet (UV) to the NIR regions, which are similar to GO (Figure 1c). The modification of DSPE-PEG-NH_2_ did not destroy the original structure of GO. The heating curves of different concentrations of GO-PEG-NH_2_ aqueous solution under 808 nm laser irradiation were measured to further evaluate the photothermal properties of GO-PEG-NH_2_. The temperatures of GO-PEG-NH_2_ aqueous solutions increased rapidly with the extending irradiation time and the increasing concentration of GO-PEG-NH_2_, as shown in Figure 1d. It is worth noting that, after 5 min. of irradiation, the temperature of GO-PEG-NH_2_ aqueous solution can be increased to 55 °C at the concentration of 50 μg/mL, which is sufficient for the bacteria to ablate. The photothermal conversion cycling experiment showed that the photothermal effect of GO-PEG-NH_2_ was almost unchanged during the heating process after five cycles of irradiation, which proved that it has good photothermal stability (Figure 1e). The infrared thermal images of 50 μg/mL GO-PEG-NH_2_ aqueous solution under 808 nm laser irradiation was collected, and the infrared thermal images were consistent with the photothermal heating curves (Figure 1f). Strong photothermal properties make GO-PEG-NH_2_ an excellent photothermal agent.

### 3.3. Selective Adsorption of GO-PEG-NH_2_

Confocal laser scanning microscopy (CLSM) was used to directly visualize the selective association of GO-PEG-NH_2_ in a mixture of live cells and bacteria to verify that GO-PEG-NH_2_ could selectively target bacteria over mammalian cells. Gram-positive bacteria (*Staphylococcus aureus*) and Gram-negative bacteria (*Escherichia coli*) were used as representative strains in order to verify the selective association of GO-PEG-NH_2_ with bacteria. At the same time, the suspension of CCRF-CEM cells was selected for experimentation because it does not require trypsin digestion to avoid undesirable autofluorescence. Firstly, we used isopropyl-β-D-thiogalactoside (IPTG) to induce *E. coli* expression of EGFP as a marker to visualize the selective binding mechanism of GO-PEG-NH_2_, and when considering the excellent fluorescence properties of enhanced green fluorescent protein (EGFP). When compared with other markers, EGFP has the advantages of small molecular weight, non-toxic effect in bacterial and eukaryotic cells, and stable fluorescence. Secondly, CLSM images of IPTG-induced *E. coli* and CCRF-CEM cells blended with GO-PEG-NH_2_ treatment were obtained to validate our strategy. In subsequent experiments, *E. coli* and CCRF-CEM cells were mixed and incubated with GO-PEG-NH_2_ at 37 °C for 30 min. The results showed that the bacteria were attached to GO-PEG-NH_2_, while most of the cells were still in a freely dispersed state (Figure 2a,e). It should be pointed out that a small number of cells were also adsorbed by GO-PEG-NH_2_. The probable reason is that part of the apoptosis exposes the phosphatidylserine (PS) in the inner leaflets of the plasma membrane, resulting in a large increase in the negative charge density on the cell membrane surface. After additional 30 min. incubation, the fluorescent image of the supernatant showed that only a small amount of bacteria was present and the cells were normally dispersed in the solution (Figure 2b,f).

IPTG-induced *E. coli* and CCRF-CEM cells were incubated with GO-PEG-NH_2_ at 37 °C for 30 min., respectively, to further determine the selectivity of GO-PEG-NH_2_ for bacteria. Clear fluorescence images indicate that GO-PEG-NH_2_ can be associated with bacteria (Figure 2c,g) without interacting with CCRF-CEM cells (Figure 2d,h). As a control, morphological images of bacteria and cells at the same concentration were collected, as shown in Appendix A, the observed cell and bacteria morphology keeps well and no characteristic changes were detected for the case of cells and bacteria untreated with GO-PEG-NH_2_. All of the above data reveal the fact that GO-PEG-NH_2_ can selectively bind to bacteria, but not to cells. The cell wall of bacteria is mainly made of peptidoglycan and acid polysaccharide, which exhibits a large amount of net negative charges on the membrane surface. In normal cells, PS is strictly confined in the inner leaflets of the plasma membrane, which results in its cell membrane being nearly neutral. Most importantly, the use of PEG to reduce protein adsorption and cell adhesion is well documented in the literature [16,53]. These factors determine that GO-PEG-NH_2_ tends to bind to bacteria, but not to cells.

The interaction mechanism between GO-PEG-NH_2_ and bacteria was analyzed by measuring the surface potential of bacteria and cells in order to further understand that GO-PEG-NH_2_ can selectively recognize and capture bacteria. Table 1 summarizes the ζ potentials of *S. aureus*, *E. coli*, and CCRF-CEM incubated with and without GO-PEG-NH_2_. The potentials of *E. coli* (from −51.83 ± 0.61 mV to −14.6 ± 0.79 mV) and *S. aureus* (from −34.63 ± 0.12 mV to −25.7 ± 0.91 mV) became more cationic after incubation with GO-PEG-NH_2_, indicating that *E. coli* and *S. aureus* successfully bind to the surface of GO-PEG-NH_2_. This is due to the active amino group on the surface of GO-PEG-NH_2_ that can act as a “linker” between the material and bacteria through electrostatic attraction. The potential of CCRF-CEM cells (from −11.70 ± 0.16 mV to −10.8 ± 0.29 mV) does not change much before and after incubation with GO-PEG-NH_2_. Bacteria carry more negative charges than the surface of the cells, which provide the most direct driving force for selective association in a mixed system, as mentioned above. Therefore, interaction between bacteria and GO-PEG-NH_2_ might be determined by the localized interaction between positively charged amino groups and negatively charged bacterial surfaces. The fluorescence image shows the fact that CCRF-CEM cells do not easily bind to GO-PEG-NH_2_ and, therefore, the ζ potentials remain almost unchanged. The results of the potential test are in line with the conclusions obtained by CLSM.

### 3.4. Antibacterial Ability

Based on the selective affinity of GO-PEG-NH_2_ to bacteria and its potential antibacterial activity, the antibacterial activity of GO-PEG-NH_2_ was subsequently studied. The MTT assay was used to analyze mammalian cells viability and the antibacterial test was evaluated with the colony counting method. Cell viability slightly decreases with the increase of GO-PEG-NH_2_ concentration, as illustrated in Figure 3a. However, cell viability still keeps at about 90%, even when the concentration reaches 70 μg/mL. That is to say, the cell viability of CCRF-CEM did not change significantly as the concentration of GO-PEG-NH_2_ increased, indicating that GO-PEG-NH_2_ had no obvious toxicity to cells. Good stability and low biotoxicity in biological media make GO-PEG-NH_2_ have great potential for biological applications. For *S. aureus* and *E. coli*, the traditional plate counting method first studied the sterilization efficiency of GO-PEG-NH2 with different concentrations. From the histogram displayed in Figure 3b, the sterilization efficiency rapidly increased with the increasing concentration of GO-PEG-NH_2_. Impressively, when the concentration of GO-PEG-NH_2_ is 30 μg/mL, the killing efficiency could reach 80%. Under the condition of 808 nm laser irradiation, the survival rate of bacteria could reach more than 90%, as shown in Figure 3c. At the same time, GO-PEG-NH_2_ also has a certain killing effect on bacteria under dark condition. This is due to the antibacterial properties of graphene oxide itself, and the positively charged GO-PEG-NH_2_ group can effectively depolarize the bacterial membrane. After the 808 nm laser was irradiated for 5 min., the temperature of the solution had risen to 55 °C, and the colony counting showed that the killing efficiency of GO-PEG-NH_2_ could reach more than 99%. Under NIR irradiation, electrons gain energy from the ground state and transition to an excited state with higher energy. Unstable electrons release thermal energy in the process of returning to a low energy state, which is the most direct cause of bacterial photothermal ablation. The gel images of bacteria and the colony count statistics of the plate counting method are given in the supporting information (Appendix A). 

Field-emission SEM was used to observe the morphological changes of bacteria in order to visually understand the damage of GO-PEG-NH_2_ to bacteria before and after 808 nm laser irradiation. For *S. aureus* and *E. coli*, a control group without GO-PEG-NH_2_, the bacteria were intact, the surface was smooth, and the edges were visible (Figure 4a,d). When GO-PEG-NH_2_ was added under the dark, the bacteria undergo a certain degree of agglomeration, and the surface of the bacterial membrane was damaged and collapsed (Figure 4b,e). This is consistent with the views stated above. After the 808 nm laser irradiation, the bacterial membrane was completely broken and the membrane was fused (Figure 4c,f). The direct information from SEM images clearly shows the damage caused by GO-PEG-NH_2_ on bacterial membrane under both dark and 808 nm laser irradiation.

## 4. Conclusions

In summary, we have reported the multifunctional photothermal material GO-PEG-NH_2_ for the selective recognition, capturing, and photothermal killing of bacteria over mammalian cells. When the cells undergo bacterial infection, the higher negative charge density on the bacterial surface provides an inherent electrostatic driving force for GO-PEG-NH_2_, which can competitively bind to bacteria, but not to cells. Therefore, the bacteria-attractive GO-PEG-NH_2_ provides a simple method for differentiating bacteria and cells. At the same time, GO-PEG-NH_2_, as an excellent photothermal agent, can effectively kill bacteria. The antimicrobial activity of GO-PEG-NH_2_ arises from two parts: the thermal energy generated by the 808 nm laser irradiation and the toxicity from positively charged amino groups to depolarize the bacterial membrane. Besides, the GO-PEG-NH_2_ is highly stable in various biological media and it exhibits low cytotoxicity, suggesting that it holds great promise for biological applications. In short, by combining the competitive bacteria adsorption property and the excellent antibacterial ability into a versatile system, this work provides new insight into graphene-based materials as a PTT agent for the development of new therapeutic platforms.

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
