# Peer review of "Graphene Oxide Composite for Selective Recognition, Capturing, Photothermal Killing of Bacteria over Mammalian Cells"

_polymers, 2020, doi:10.3390/polym12051116_

Round 1

Reviewer 1 Report

Manuscript ID: polymers-783951

This submission indeed may become suitable for publication in polymers if authors could address the following comments carefully.

  • Authors have to provide clear evidences to ascertain the formation of GO-PEG-NH2 by means of IR and NMR.
  • What is the loading of PEG-NH2? Does it have any influence on its activity?
  • Authors should also comment briefly on the applications of GO and its functionalized solids in catalysis by citing seminal contributions from H. Garcia, D. Su and P.Serps research groups.
  • Authors should try to polish the language and typographical errors have to be corrected.

Reviewer 2 Report

The study entitled “Graphene Oxide Composite for Selective 2
Recognition, Capturing, Photothermal Killing of 3 Bacteria Over
Mammalian Cells Kaolin alleviates the toxicity of graphene oxide for
mammal cells” provides a very interesting enhanced antibacterial
approach in the current antimicrobial resistant era. However, several
points should be addressed in order to be suitable for publication:

1) The introduction section must be improved. Thus, after the first
sentence (line 32) which is very important nowadays due to microbial
resistance, it should be mention that graphene oxide and other carbon
nanomaterials have shown to be very promising nanoweapons against
multidrug-resistant bacteria with references such as [European Polymer
Journal,  110, 14-21 (2019),
https://doi.org/10.1016/j.eurpolymj.2018.11.012; Polymers 2019, 11(3),
453, https://doi.org/10.3390/polym11030453;]. It should be mention also
that the antibacterial properties of carbon nanomaterials against
multidrug-resistant pathogens can be enhanced by absorbing light irradiation  and provide some references from literature.
2) It would be better to mention in subsection 2.1 the product code of
the graphene oxide used in this study for future reproducible results.

3) In materials and methods, in line 142, a reference should be given for
the colony counting method typically used for the antimicrobial characterization of advanced materials. Another reference should be given for the
sample preparation method followed for the SEM observation

4) In line 154 the word measurements should be removed because no
measurements were performed by SEM observation

5) In line 204-206, the authors should explain better how the manage to
produce rounded edges

6) In line 275 the sentence “Most importantly, the use of PEG to reduce
protein adsorption and 275 cell adhesion is well-documented in the
literature.” Should go with some references at the end

7) Information such as irradiation time and bacterial culture time
should be included in the legend of figure S4 and Tables S1, S2 and S3

Round 2

Reviewer 1 Report

The revised version is now may be considered for publication in Polymers.

Reviewer 2 Report

The authors have improved the manuscript as suggested and now it is ready for publication.